

# Review of intervention methods for language and communication disorders in children with autism spectrum disorders

Mengmeng Cui[1], Qingbin Ni[2] and Qian Wang[2]

[1] Department of Rehabilitation, The Second Affiliated Hospital of Shandong First Medical University, Taian, Shandong, China
[2] The Affiliated Taian City Central Hospital of Qingdao University, Taian, Shandong, China

## ABSTRACT

In recent years, the number of patients—particularly children—with autism spectrum disorder (ASD) has been continually increasing. ASD affects a child's language communication and social interaction to a certain extent and has an impact on behavior, intelligence level, and other aspects of the child. Data indicates that 40% to 70% of children with ASD experience language developmental delays, which are mainly manifested as lack of language or language developmental delay, self-talk, use of stereotyped language, parroting, et cetera. A language communication disorder is a major symptom of ASD and is the most common reason for patients to visit a doctor. Therefore, language intervention training and communication skills have been made a cornerstone of autism intervention. However, a literature search has revealed that most studies only examine certain intervention methods or a combination of two or three intervention methods, which cannot be used by therapists or rehabilitation teachers. Therefore, this article summarizes relevant literature on language communication training for ASD children at home and abroad and briefly introduces the characteristics and training methods of language disorders in children with ASD in order to provide some ideas and references for relevant researchers and practitioners.

## INTRODUCTION

Autism spectrum disorder (ASD) is a neurodevelopmental disorder. It usually begins before the age of 3 years and is characterized by persistent social communication impairment and loss of social interaction, repetition of stereotyped language, or behavior. The latest edition of the Diagnostic and Statistical Manual of Mental Disorders (DSM-V) has eliminated the classification of clinical subtypes according to the diagnostic criteria and the classification of ASD as outlined in DSM-IV. Instead, three categories of disorders, namely autistic disorder, Asperger's syndrome, and pervasive developmental disorder not classified, are collectively referred to as ASD (*Hantsoo et al., 2019*). In 1990, the prevalence of ASD was approximately 7.5/1000, and the number of confirmed cases has now reached more than 10 million. According to a report released by the US Centers for Disease Control and

Corresponding authors
Qingbin Ni, ni_qingbin@163.com
Qian Wang, qianqian-wangxi@163.com

Prevention in 2020, the incidence of autism continues to rise. ASD prevalence has changed from 1/59 in 2018 to 1/54. The incidence of autism in China is as high as 0.7%, and there are currently 10 million ASD patients, including an estimated 3 million children under the age of 12. The male-to-female ASD ratio is approximately 4–5:1 (*Hayward et al., 2009*). The etiology of ASD is unclear, and its increasing incidence has prompted researchers to pay attention to the affected population.

As with most disease treatments, language intervention for children with ASD can be divided into symptomatic treatment and targeted treatment. The cause of ASD is unclear, but the effectiveness of symptomatic treatment has been gradually confirmed and rehabilitation training or rehabilitation education has been advocated. Early family intervention is urgent and important. There are many existing language intervention methods, but most of them only present findings from individual cases or small sample groups. The challenge lies in accurately identifying specific problems faced by children and formulating effective methods tailored for different individuals. A viable comprehensive training program is a top priority.

This article examines the characteristics of language and communication disorders in children with ASD, provides an overview of existing intervention methods, highlights shortcomings and limitations of these methods, and suggests directions for future development based on practical challenges. An incomplete understanding of language and communication disorders among patients with ASD has led to variable intervention outcomes, necessitating this literature review. This literature review will be of interest to pediatricians, speech therapists, special educators, and parents involved in ASD interventions.

## SEARCH STRATEGY

We conducted a literature review using PubMed and Google Scholar to identify relevant studies. The search terms used were "autism spectrum disorder (ASD)", "autism spectrum condition (ASC)", "autism spectrum", and "autism". In addition, author names and reference lists were utilized to search for relevant references. The search was performed by author Mengmeng Cui.

## LANGUAGE INTERVENTION FOR NON-VERBAL CHILDREN

Nonverbal language intervention for young children in the non-verbal period is mostly concentrated in the one-to-three-year-old range. Nevertheless, there are older children with severe ASD and who have no oral expression. During early stages, children with autism share similarities in language expression with children with language development delay and language impairment, which is in line with the prediction of the "multidimensional consensus hypothesis". However, children with ASD have heterogeneity in language expression, including specific defect phenomena within some language categories (*Chan & Leung, 2022*).

During the non-verbal period, most children, as they increase in age, either do not speak, or the phenomenon of language regression occurs. The main goals for children

in this period are to establish basic communication awareness, improve communication attitude, and induce pronunciation. It is beneficial to refer to the assessment and training methods for language developmental delay.

Natural developmental behavior intervention (NDBI), a combination of interventions implemented in a natural environment, utilizes random emergencies and uses a variety of behavioral strategies to help children with autism develop appropriate and essential survival skills. NDBI is naturally implemented in everyday life (*Schreibman et al., 2015*). The core deficits of children with ASD are social behavior and communication impairment. The main social scene of children under 3 years old is in the daily activities of their families, and their main social interaction objects are parents and their families. Therefore, caregivers have ample opportunities to use NDBI strategies to promote the development of their children and offspring within their everyday environment (*Crank et al., 2021*; *Volkmar et al., 2014*), which also alleviates the impact of training interruptions during an outbreak to a certain extent. The "Expert Consensus on Family Interventions for Children with ASD in China" pointed out that a number of domestic and foreign studies have shown the effectiveness of ASD family interventions. These interventions have proven to be highly beneficial. Parents need to respond in a timely manner to all communication behaviors of their children, whether they are correct or inappropriate, and encourage their active communication behaviors. Responses should use language and gestures that are simple, clear, and consistent with the child's comprehension. Adaptive ability is the comprehensive use of children's cognitive, language, motor, communication, perceptual, emotional, and other abilities in daily life learning. Adaptive skills refer to skills—such as the ability to play sports, game entertainment, labor, and self-care—that can improve the adaptive level of children (*Subspecialty Group of Developmental and Behavioral Pediatrics, the Society of Pediatrics, Chinese Medical Association; Subspecialty Group of Child Health Care, the Society of Pediatrics, Chinese Medical Doctor Association, 2022*).

A randomized controlled trial (RCT) of a communication-focused intervention in children (under 12 years of age) with a diagnosis of ASD and minimal verbal ability (less than 30 function words or inability to communicate using speech alone), was compared with no treatment, wait-list control, and usual care. The RCT found that children with lower expressive abilities at baseline (age-equivalent to less than 11.3 months) progressed more than it did among children with higher expressive abilities. Some children in the intervention group showed an improvement in their expressive abilities (*Brignell et al., 2018*). This finding suggests the importance and effectiveness of early language communication intervention.

To implement early intervention for children in the early stages of language development, Lu Haijun's article "Language Rehabilitation Training for Children with Autism", suggests the following pre-language training methods for ASD children: (1) Stabilize emotions and establish basic learning behaviors. (2) Improve imitation ability and build learning ability. (3) Create a structured environment to establish correct study habits (*Yan et al., 2021*). These methods provide a basic idea of pre-language intervention.

## LANGUAGE INTERVENTION FOR CHILDREN WITH SPEECH IMPAIRMENT

Children with ASD have speech impairments: most of them have purposive dyspraxia rather than lesions of the articulatory organs. Interventions for speech impairments mainly train children's imitation ability. This training can start from the imitation of gross movements to the imitation of oral movements and fine movements, focusing on the expansion from "characters" to "words" to "short sentences" and intervention using various learning methods. The pronunciation imitation should go from silent imitation to producing sounds, starting with syllables that can be pronounced and gradually increasing the number of syllables to words and sentences. In this process of pronunciation imitation, intuitive games can be used to attract children's interest and exaggerated responses can be used to give evaluations, immediate rewards, *et cetera* so that children can pass the non-speaking period as soon as possible in the game. This approach will foster their language development.

## INTERVENTIONS FOR CHILDREN WITH COMPREHENSION IMPAIRMENTS

Comprehension ability is an important prerequisite for the development of language communication skills. At present, it is known that ASD patients are clearly backward or have defects in sharing ability, empathy, joint attention, understanding others' wishes, and cooperative behavior. These shortcomings hinder the patients' comprehension of the purpose, intention, suggestion, belief, wish, cover-up, joke, deception, comparison, "duplicity", and other psychological activities of the other party's behavior. Consequently, ASD patients cannot understand "subtexts" and "unspoken rules", and neither do they understand humor. This lack of understanding is harmful and hinders the making and execution of plans.

In children with ASD, social knowledge/behavior deficits due to difficulties in social/emotional cue detection, encoding, and interpretation lead to problems in joining and initiating social interactions. To address these difficulties, interventions can target the six steps of the social information processing model of ASD proposed by Crick and Dodge (*Torres, Saldaña & Rodríguez-Ortiz, 2016*). This model describes a series of covert psychological mechanisms for translating external social cues (inputs) into overt behavioral responses (output). It provides a theoretical framework for a clear understanding of children's socio-cognitive abilities and social adaptation. The model consists of five cognitive steps, culminating in a sixth step of behavioral response formulation: (1) encoding of internal and external social cues, (2) understanding and mentally representing cues, (3) articulating or selecting goals, (4) response structure, (5) making coping decisions, and (6) formulating behavior (*Torres, Saldaña & Rodríguez-Ortiz, 2016*; *Ziv et al., 2014*). Each step is guided by biologically determined cognitive functional abilities and a learned social experience memory "database" that provides knowledge or skills about social rules, patterns, and social behavior (*Chan & Leung, 2022*). It is possible to evaluate problem links and select intervention strategies and materials based on this six-step model theory. Poor

comprehension ability is an obstacle to the coding and recognition of internal and external social cues in the first step.

The development of comprehension is based on normal vision and hearing. Some children with ASD have abnormal auditory development. They often respond inappropriately to sounds, which may be manifested as hypersensitivity or sluggishness, and cannot correctly understand the meaning of verbal and non-verbal expressions. To address these challenges, a child should first be allowed to learn to listen. One approach is to introduce different sounds, such as animal sounds, sounds of nature, or music. Therapists and parents can speak to the children's interests: exaggerated gestures and expressions may be employed to get the child's attention and facilitate its responsiveness.

In addition, visual and auditory training for children can improve abnormal language communication in children with ASD. The picture exchange communication system (PECS) is a rehabilitation training method with many clinical applications. By using picture exchange as an auxiliary way to communicate with children, PECS can enhance children's visual tracking and understanding through structured teaching and behavior analysis. The PECS method of analysis stimulates the improvement of children's communication and language skills. A study has found that PECS causes some improvement in communication and expression, but the improvement in the level of development has not reached clinical expectations. Another approach is auditory integration training, which stimulates children's hearing through the use of a "processed" sound by adjusting the auditory tester and modifying the volume and audio frequency to suit the children. By prompting children to alter their perception of sound, AIT facilitates changes in cognitive and behavioral habits (*Schreibman et al., 2015*). There is also a method that is easy for children to accept: music therapy. Music therapy is a planned and purposeful interactive intervention method, and the corresponding music behavior or music experience is specially designed according to individual needs. Music therapy can be used for intervention training in ASD patients. Several current forms of intervention include receptive music intervention, creative music intervention, improvisational music intervention, recreational music intervention, and music games (*Weisblatt et al., 2019*).

Another study found that, compared with normal children, children with high-functioning autism had a reduced ability to distinguish language information in Mandarin Chinese in a noisy environment, indicating that children with ASD may have language perception dysfunction. To address language perception dysfunction in high-functioning autistic children, the application of family emotional training and interaction with pets can improve their sharing behavior to a certain extent; however, the initiative, autonomy, and independence of these children remain poor (*Ziv et al., 2014*; *Weisblatt et al., 2019*). An in-depth study of neuropsychology may provide an important basis for the correction of these difficulties.

## ADVANCED TRAINING CONTENT FOR CHILDREN WITH ASD

The focus of this stage is to improve the quality of learning and skills by focusing on the improvement of children's sociality. When a child has developed some awareness of

communication, the improvement of sociality should be made the focus of training. Social communication, in the language rehabilitation of ASD children, should be strengthened mainly through speech and exercise. Talking refers to the training of trainers and autistic children through chatting. This training can be carried out using everyday common-sense knowledge as content. In addition, social practice is essential. During the rehabilitation of a child, preparation for active communication with parents should be made, and social practice should be incorporated to address the child's social survival issues. For instance, parents can take their children to the mall for shopping, a visit to the pharmacy to buy medicine, or go to the bank to withdraw money. After long-term training, children with autism can achieve different degrees of social progress, which lays a foundation for the development of social interaction and survival skills in children with ASD (*Yan et al., 2021*).

## BEHAVIORAL AND PSYCHOLOGICAL INTERVENTIONS FOR CHILDREN WITH ASD

### Structured education and early psychological intervention

This study found that structured education combined with early psychological intervention can improve the clinical symptoms of children with ASD and promote their recovery. Structured education requires setting up a working system tailored to the situation of the child, which includes organizing the environment, implementing visual supports, and establishing routines and schedules. This schedule allows a child to understand and master skills on what to do, how to do it, and, as far as possible, create time limits for tasks, which will enable it to cultivate the ability to work independently. Psychological intervention played a role in the treatment process: parental guidance was emphasized and the ''situational trigger'' method was followed to guide the child's engagement in interpersonal interactions. Parents were encouraged to take the child outdoors and to facilitate its participation in group activities with children of the same age. Psychological interventions, for example, making statements such as ''Look, how fun this is; can we play with this child? Guess what he is doing'', were utilized to guide children to think independently. When there were enough participants, group activities such as music and group games were encouraged to prevent independent play and respond actively when children had needs (*Zeng et al., 2021*).

### Applied behavior analysis (ABA)

ABA is a set of training techniques and operating systems for autism proposed by American psychology professor Ivar Lovaas. ABA is also known as reinforcement therapy. It has been widely used around the world as an intervention for children with autism. One method within ABA is discrete trial teaching (DTT), which was proposed by Lovaas and Koegel. DTT is also known as the discrete unit teaching method and can only raise approximately 50% of the abilities of children on whom an intervention has been done close to normal children's level. DTT cannot solve some problem behaviors. Therefore, Lovass and Koegel proposed a key skill training method (pivotal response training; PRT) that focuses on technical means training. Since the 1960s, extensive international research has been done on the application of DTT and PRT to the behavioral problem intervention of

autistic children. The research has established many findings on interpersonal interaction, language communication, behavioral cognition, and emotion. Scholars in Taiwan have also conducted empirical research, which has revealed that the DTT intervention model can help children with autism to cultivate and learn behavioral goals to a certain extent. However, the main purpose of the PRT intervention model is to change the core behavior of children with autism through positive training, thereby enhancing their communication skills, game skills, social interaction behavior, and self-control and to develop key skills to achieve integrated developmental goals (*Jobin, 2020*). Nevertheless, PRT is reported to be rejected and feared by ASD children who have received ABA training. Some scholars believe that positive reinforcement is a kind of "conditional love" and a serious weakening of self-evaluation (*Sitoiu & Panisoara, 2021*). This viewpoint deserves thoughtful consideration. In recent years, some researchers have tried combined interventions such as ABA combined with childlike intervention and ABA combined with structured teaching, namely treatment and education of autistic and related communication-handicapped children, and obtained satisfactory results (*Gould, Tarbox & Coyne, 2018*; *Callahan et al., 2010*).

# EMOTIONAL MANAGEMENT IN CHILDREN WITH ASD

Children with ASD often have difficulty regulating emotions, posing difficulties in implementing various interventions when emotional problems arise. Therefore, it is particularly important to deal with the emotional problems of children with ASD. Research has shown that children's emotional outbursts and reduced use of passive comfort strategies are associated with a low quality of life at home. Many parents of children with autism observe that their emotions can influence their child's mood and behavior, and vice versa, a phenomenon known as "emotional transmission". Children who often do not actively seek to change their environment, but passively accept it, have a difficult time functioning in a family and achieving a positive quality of life. In the families of children with ASD, passive comfort strategies have a negative relationship with family well-being. Therefore, teaching other aggressive strategies should be encouraged. These positive strategies are recommended in place of negative comfort strategies. Additional research is needed on strategies for emotion regulation (*Nuske et al., 2018*).

Studies have found that 12%–20% of autistic patients will develop catatonia and have reported the effectiveness of electroconvulsive therapy (*Wachtel, 2019*). Catatonia is currently thought to be a periodic-course syndrome characterized by changes in movement, speech, and behavior that typically occurs in the context of a variety of somatic and neuropsychiatric disorders. There are few case reports and systematic reviews of ASD complicated with catatonia in China, indicating that this important clinical problem is still seriously neglected. Some researchers have diagnosed and treated 13 ASD children with catatonia. After identifying the source of stress and adverse ecological environment, parents were guided to increase their companionship and care for their children, learn and change their educational methods, and adhere to the principles of ASD intervention education and the comprehensive behavior–structured–social intervention model so as to foster understanding, tolerance, and acceptance. Beating or scolding

was resolutely avoided. The parents' education methods were unified, the daily life of the children was organized reasonably, and efforts were made to create a calm, stable, regular, and happy living environment for the children. In addition, the children are given appropriate, low-dose standardized medication. Consequently, the symptoms of catatonia were significantly improved. This study suggests that ASD children with catatonia may be likely to be interpreted as having "catatonia-like episodes", which are severe disruptions of the homeostasis state after experiencing a harsh ecological environment. When a friendly living environment was provided and when scientific and effective interventions were made, the behavior and functioning of the children gradually returned to a stable homeostasis state (*Li & Zou, 2021*).

## INTERVENTIONS FOR INTELLECTUAL DISABILITY

Regarding the intellectual characteristics of children with ASD, some scholars initially believed that the intelligence of individuals with autism was normal. However, later studies showed that approximately three-quarters of the children had an intellectual disability. In recent studies conducted in Sweden and Canada, the rate of intellectual disability in typical autism has been found to be 80% to 89% and 76%, respectively. An estimated 40% to 60% of children have an IQ of less than 50 points, and 20% to 30% have an IQ of less than 50. Most children with ASD who have a score of 70 or higher exhibit inattention or are unable to respond verbally; besides, they cannot interact flexibly. Language barrier further affects the interaction of these children with people, thereby aggravating the social barriers of the children (*Tawhid et al., 2021*).

Giving speech training only to these children has a poor effect on improving their overall ability. Studies have shown that sensory integration training can stimulate various parts of the body with different modalities, including audio-visual, tactile, and proprioceptive systems to correct and stimulate the ability of children to make normal responses to promote the recovery of their nervous system and improve their intelligence (*Iwanaga et al., 2014*). In addition, early cognitive response training for children is an important approach to helping ASD children with intellectual disability. There are studies reporting that hyperbaric oxygen therapy is effective in children with ASD, but further, standardized application and clinical validation are needed to establish the efficacy of the therapy.

## OTHER INTERVENTION METHODS

### Inclusive education

Inclusive education advocates that children with special needs and healthy children should enjoy formal education together in a general education environment. At present, countries around the world are actively advocating for inclusive education. A number of studies have shown that inclusive education provides children with autism with a natural environment and rich stimulation and can effectively promote the social development of children with autism (*Hess et al., 2008*). Recently, a study plan released by the education department has also gradually begun to be implemented. This plan will definitely solve the education

problems of some children. This development is seen as a milestone by parents of children with ASD.

### Interest-oriented floor game

Interest-oriented floor games stimulate and attract children, encouraging them to actively communicate with others through interactive games, which can enrich and improve their ideas and thinking and emotional development (*Wieder, 2021*). An interest-oriented floor game uses a child's interest as a guide and changes from a passive to an active approach. Within a relaxed and entertaining atmosphere, the therapist adopts purposeful guidance based on observations and interactions with the child, thus arousing its desire for communication and fostering the development of its social language, innovative thinking, and other abilities (*Amaral et al., 2018*).

### Intervention method of traditional Chinese medicine theory for language communication disorders in children with ASD

With respect to children with ASD, acupuncture and moxibustion have been researched in traditional Chinese medicine. Children with ASD are likely to suffer from heart and liver embolisms (*Bang et al., 2017*).

Inflammatory cytokines affect the brain and play an important role in the development of autism (*Hantsoo et al., 2019*). Animal experiments have shown that Zhisan acupuncture can inhibit apoptosis in the hippocampus, thereby improving the learning and memory function of rats (*Hou et al., 2015*) and promoting the recovery of their cognitive functions (*Wei, Xie & Yuan, 2019*).

There are also studies that employ Sishen acupuncture, temporal three acupuncture, wisdom three acupuncture, and acupuncture on speech areas 1 to 3 based on the characteristics of human brain function divisions. However, the "Expert Consensus on Early Intervention in Families of Infants and Young Children with ASD" mentions the lack of convincing worldwide evidence for the effectiveness of acupuncture and moxibustion in the treatment of ASD.

### Drug intervention

The etiology of ASD is unknown, and there is currently no specific drug treatment for its core symptoms (*Magaña, Lopez & Machalicek, 2017*). Instead, there are medications that help relieve some symptoms but should only be used when behavioral problems are prominent and other treatments are ineffective. Further research is needed to determine the use of medications.

### Training for ASD caregivers

Caregivers of individuals with ASD play a vital role in supporting the recovery of ASD patients by providing a safe home, proper nutrition, and enriching educational opportunities. Research suggests that caregivers play a key role in the development of individuals with ASD, and there is a need for a personalized medical approach to tailor parental intervention strategies (*Swanson, 2020*).

## EMERGING TECHNOLOGIES

### Virtual reality (VR) technology

Emerging VR technologies are used as educational and intervention tools for people with ASD. Based on mainstream rehabilitation education theories, VR-based platforms and devices offer advantages in the cultivation of social communication and interaction skills. Evidence-based practice shows that incorporating VR into treatment or training programs is an effective way of improving social performance in people with ASD. After a VR-based intervention, individuals with ASD have shown significant improvements in social functioning, emotion recognition, speech, and language abilities (*Zhang et al., 2022*).

### Social assistance robot

Studies have revealed that social assistance robots seem to have the potential to be an important collaborative tool in recreational therapy for children with ASD, allowing them to experience additional physical and mental interactions and great participation in therapy (*Panceri et al., 2021*). Ongoing research and development and design of relevant games and functions are expected to have social, therapeutic, and scientific relevance. This research will update and optimize care for children with ASD.

### Wearable technology (WAT)

Assistive technology for social skills learning, especially WAT, has been a popular rehabilitation research topic for individuals with ASD in recent years. With the help of WAT, people with autism not only learn in a variety of settings but also have real-time feedback in a real social environment. These advances are an important means of increasing the chances of children with ASD being integrated into society and deepening their knowledge and awareness of real social situations. Rather than simply thinking about what technology can do, WAT design considers the needs or preferences of individuals with ASD. As WAT matures, it improves the confidence of families that their members who have ASD will recover (*Koumpouros & Kafazis, 2019*; *Benssassi et al., 2018*).

## CONCLUSION

This article only provides a general and comprehensive introduction to the language communication intervention methods for children with ASD, which can provide a general framework for practical application by ASD rehabilitation therapists and teachers. A large number of intervention methods have been proven effective and have been applied in clinical practice. However, some methods have only been validated in individual cases or small sample groups. Each child with ASD has different symptoms and problems that require unique intervention. Intervention using language communication should not be handled singularly: training or treatment requires a comprehensive, multi-angle intervention approach. To rehabilitation therapists and teachers of children with ASD, each child is a different challenge; therefore, they must continuously improve their "training material library" and master the indications and corresponding intervention methods and develop the ability to quickly adjust their training plans in the face of new emerging problems.

For clinicians and researchers, discovering and validating the pathogenic factors behind ASD in children is an urgent problem that should be addressed. Doing so would be good news to many families, who are affected by this disease, around the globe. Given the trend of annually increasing incidence of ASD and the present absence of a known cause or specific medication, the intervention environment based on hospitals or rehabilitation institutions has been unable to meet the great current demand (*Zwaigenbaum et al., 2015*). Therefore, it is crucial for society to accommodate and respect individuals affected by ASD and adopt the mainstream advocated model of combining medicine and educators to jointly face and intervene in various problems of children with ASD. This approach entails the best possible exploration of the potential of these individuals who have been ruined by ASD so that they can realize their value in society. Achieving this goal requires the collective efforts of our entire society.

### Funding

The study was funded by grants from the Shandong Medical and Health Technology Development Fund (202103070325), Shandong Province Traditional Chinese Medicine Science and Technology Project (M-2022216) and Nursery Project of Taian City Central Hospital (2022MPM06). The funders had no role in study design, data collection and analysis, decision to publish, or preparation of the manuscript.

### Grant Disclosures

The following grant information was disclosed by the authors:
Shandong Medical and Health Technology Development Fund: 202103070325.
Shandong Province Traditional Chinese Medicine Science and Technology Project: M-2022216.
Nursery Project of Taian City Central Hospital: 2022MPM06.

### Competing Interests

The authors declare there are no competing interests.

### Author Contributions

- Mengmeng Cui conceived and designed the experiments, performed the experiments, analyzed the data, prepared figures and/or tables, authored or reviewed drafts of the article, and approved the final draft.
- Qingbin Ni conceived and designed the experiments, performed the experiments, analyzed the data, prepared figures and/or tables, and approved the final draft.
- Qian Wang conceived and designed the experiments, performed the experiments, analyzed the data, prepared figures and/or tables, and approved the final draft.

### Data Availability

This is a literature review.

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
