# Peer review of "(untitled)"

_PeerJ, doi:10.7717/peerj.15735_

## Round 0.1 · original submission · Minor Revisions

Please carefully read the comments and suggestions from the editors and reviewers and provide a point-by-point response.

Reviewer 1 ·

Basic reporting

The prevalence of ASD is increasing every year and has become a public health problem. Language impairment is one of the core symptoms of ASD. This article reviews the progress of research on language interventions in autism spectrum disorders and is a practice-oriented article.

Experimental design

None.

Validity of the findings

The authors' mentioned interventions for language and social impairment in ASD are comprehensive, and the findings of their review are valid for clinical prevention and treatment of language impairment in ASD.

Additional comments

ASD has both full and abbreviated forms in the article, and it is recommended to standardize the writing criteria. The general rule is to give the full name and abbreviation of the ASD when it first appears, followed by the abbreviation.

Reviewer 2 ·

Basic reporting

ASD is a lifelong developmental condition that affects the way people perceive the world and interact with others. Typical social engagement challenges, which are common in the experience of individuals with autism, can have a significant negative impact on the quality of life of individuals and families with autism. This article reviews the current state of research on language interventions for autism, which has important theoretical and practical implications.

Experimental design

No comments.

Validity of the findings

This article provides only a general and comprehensive introduction to verbal communication intervention methods for children with ASD, which can provide a general framework for practical application by ASD rehabilitation therapists and rehabilitation teachers. However, intervention methods still need to be used in clinical practice to prove their effectiveness.

Additional comments

The manuscript language needs to be further embellished to make it more readable.

Reviewer 3 ·

Basic reporting

This is a comprehensive review that summarizes the language disorders of ASD.

Experimental design

It is a comprehensive review.

Validity of the findings

The review describes recent advances on language disorders in ASD, with accurate citations to the literature and no obvious errors.

Additional comments

1. The heading "Search Methodology" should be changed to "Search strategy".
2. I think the previous literature review with the same research topic should also be cited appropriately.
3. I think a paragraph talking about the future outlook should be supplemented, which should contain the research gap on this topic that needs to be urgently addressed for future researchers.
4. I think "12. summarize" should be replaced by "conclusion".
5. I think this manuscript should be carefully proofread by a fluent English speaker in the neurology field.

---

## Round 0.2 · accepted · Accept

The paper has been well-revised and is acceptable for publication.